# The PIP Peptide of INFLORESCENCE DEFICIENT IN ABSCISSION Enhances *Populus* Leaf and *Elaeis guineensis* Fruit Abscission

**DOI:** 10.3390/plants8060143

**Published:** 2019-05-30

**Authors:** Timothy John Tranbarger, Hubert Domonhédo, Michel Cazemajor, Carole Dubreuil, Urs Fischer, Fabienne Morcillo

**Affiliations:** 1UMR DIADE, Institut de Recherche pour le Développement, Université de Montpellier, 34394 Montpellier, France; 2Ecology and Genetics Laboratory, Pontificia Universidad Católica del Ecuador (PUCE), 17-01-21-84 Quito, Ecuador; 3CRAPP, INRAB, BP 1 Pobè, Benin; hubertdomonhedo@yahoo.fr (H.D.); Michel.Cazemajor@palmelit.com (M.C.); 4PalmElit SAS, F-34980 Montferrier-sur-Lez, France; 5Department of Forest Genetics and Plant Physiology, Umeå Plant Science Centre, Swedish University of Agricultural Sciences, SE-901 83 Umeå, Sweden; Carole.DUBREUIL@cea.fr (C.D.); Urs.Fischer@kws.com (U.F.); 6DRT DPACA, CEA Tech Cadarache, 13108 Saint Paul Lez Durance, France; 7KWS SAAT SE, RD-BT, 37574 Einbeck, Germany; 8UMR DIADE, CIRAD, F-34398 Montpellier, France; fabienne.morcillo@ird.fr

**Keywords:** organ abscission, fruit abscission, leaf abscission, cell separation, peptide signalling, LRR-RLK, *Populus*, oil palm, abscission zone

## Abstract

The programmed loss of a plant organ is called abscission, which is an important cell separation process that occurs with different organs throughout the life of a plant. The use of floral organ abscission in *Arabidopsis thaliana* as a model has allowed greater understanding of the complexities of organ abscission, but whether the regulatory pathways are conserved throughout the plant kingdom and for all organ abscission types is unknown. One important pathway that has attracted much attention involves a peptide ligand-receptor signalling system that consists of the secreted peptide IDA (INFLORESCENCE DEFICIENT IN ABSCISSION) and at least two leucine-rich repeat (LRR) receptor-like kinases (RLK), HAESA (HAE) and HAESA-LIKE2 (HSL2). In the current study we examine the bioactive potential of IDA peptides in two different abscission processes, leaf abscission in *Populus* and ripe fruit abscission in oil palm, and find in both cases treatment with IDA peptides enhances cell separation and abscission of both organ types. Our results provide evidence to suggest that the IDA–HAE–HSL2 pathway is conserved and functions in these phylogenetically divergent dicot and monocot species during both leaf and fruit abscission, respectively.

## 1. Introduction

Organ abscission is a process that enables plants to shed whole organs that have completed their function or as a defence response when infected or damaged [1]. At the molecular level, organ abscission is a complex process that includes the integration of both endogenous and exogenous signals derived from the physiological and environmental status of the plant [2]. At the cellular level, organ abscission is dependent on mechanisms that cause adjacent cells to lose their adherence to one another and separate [1]. Furthermore, these cell separation mechanisms involve cell wall modifications, in particular the hydrolysis of pectin within the middle lamella of adjacent cells, which occur specifically with cells at the base of the organ to be shed, within a specialized tissue called the abscission zone (AZ) [1,3]. The function of the AZ allows plants to shed almost any organ, including leaves, flowers, floral parts, and fruit at the appropriate time. While these various organ types have different ontogenies, with few exceptions, the basis for organ abscission is related to the function and signalling mechanisms that occur in the AZ.

Molecular genetic research on floral organ abscission with the model species *Arabidopsis thaliana* has led to tremendous progress in deciphering the underlying mechanisms and to the definition of details concerning the stages of organ abscission [4,5,6,7,8]. While these signalling and molecular pathways identified in *Arabidopsis* provide insight into the stages and signalling mechanisms of organ abscission, it is not known whether this information can be translated to all species and all organ types that are shed [9]. Based on the current model derived from these studies, the four stages of abscission include: (1) the ontogeny of the AZ, (2) the acquisition of AZ cell competence to integrate and respond to abscission signals, (3) the activation of AZ cells by abscission signals that induce the de novo synthesis of cell wall-modifying enzymes that cause cell wall loosening and rounding of the AZ cells, and (4) the trans-differentiation of remaining AZ cells to produce a protective layer [8,10].

One signalling pathway discovered in *Arabidopsis*, which has become a significant focus of research, involves a peptide ligand-receptor system [10,11,12]. The signalling pathway consists of the secreted peptide IDA (INFLORESCENCE DEFICIENT IN ABSCISSION) encoded by the gene *INFLORESCENCE DEFICIENT IN ABSCISSION* (*At1g68765*), and the two leucine-rich repeat (LRR) receptor-like kinases (RLK) HAESA (HAE) and HAESA-LIKE2 (HSL2) [12,13,14]. It was demonstrated that the IDA peptide binds and activates HSL2, and that the 12 amino acid long PIP peptide with hydroxylation of a central proline residue is the most efficient peptide for activation [13]. Both *ida* and *hae hsl2* mutants retain their floral organs, while the overexpression of the *IDA* gene rescues the *ida* mutant, which reverts to the wild-type abscission phenotype, and overexpression in the wild-type of either *IDA* or similar *IDA-LIKE (IDL)* gene family members results in early abscission [11,12]. A recent survey of the literature suggests that the IDA–HAE–HSL2 pathway functions during the final stages of organ abscission (stages 3 and/or 4), and not during the stages of acquisition of competence or activation of the AZ (stages 1 and 2) [10]. However, while it is clear that the IDA–HAE–HSL2 signalling pathway functions during *Arabidopsis* floral organ abscission, it is not known whether this pathway functions in other species and with organs such as leaves or fruit.

Several recent studies have begun to respond to the question of whether the IDA–HAE–HSL2 pathway functions in different species and organs. For example, *IDL* genes identified in both citrus and litchi ectopically expressed in *Arabidopsis* promoted floral organ shedding, suggesting a conserved role for this pathway in both these species [15,16]. While these studies show conservation of function in the context of *Arabidopsis*, evidence of function within citrus and litchi is not available. A previous study by our groups identified similar genes that encode IDL, HAE, and HSL2 proteins from *Populus* and oil palm (*Elaeis guineensis*) with expression correlated to leaf and fruit abscission, respectively [17]. Both of these organ abscission processes are important for the production capacities of each of these species; fruit yield and the facilitation of harvest for oil palm and biomass production for poplar. While our previous studies showed that the functional components of the pathway are present, no evidence is available on the function of the pathway within these two species. In the current study, we address the question of whether the PIP peptide of IDA has the capacity to enhance these different organ abscission processes in two widely divergent species. The results with both organ abscission systems showed that small but significant increases in the abscission of both leaf and fruit occur after treatment with the PIP peptide of IDA. The results demonstrate the bioactive capacity of the PIP peptide of IDA in both these abscission systems, and suggest that the IDA–HAE–HSL2 pathway is conserved and functions in these phylogenetically divergent dicot and monocot species during both leaf and fruit abscission, respectively.

## 2. Results

### 2.1. Exogenous Treatments of *PtIDA* and *PtIDL1* Peptides and Their Effect on Poplar Leaf Abscission

Leaf abscission in the hybrid aspen T89 (*Populus tremula* X *P. tremuloides*) is induced by dark treatment of leaves, which has been used previously as an experimental system to analyse the abscission process at the molecular level [17,18]. Using the same experimental system, it was previously shown that the *Populus PtIDA* and *PtIDL1* gene transcripts increased significantly during dark induced leaf abscission [17]. In the current experiments, this system was used to test whether either the PtIDA or PtIDL1 synthetic peptides have a bioactive effect during dark-induced leaf abscission in *Populus*. Our hypothesis was that these peptides might be able to enhance leaf abscission after dark treatment, which would provide further evidence that the IDA–HAE–HSL2 pathway functions during dark-induced leaf abscission of *Populus*. For the experiment, at the same time as the dark induction of leaf abscission (t = 0), leaf axils were treated with either PtIDA or PtIDL1 synthetic peptides at the same time (t = 0), and again at 7 days after dark induction (DAD). After 1 day of the second peptide treatment (8 DAD), 10% more leaves shed from plants treated with either PtIDA or PtIDL1 (Figure 1a). A higher percentage of leaf abscission continued to be observed under both peptide treatments up until 12 DAD, while no increase in percentage of leaves shed was observed at the remaining time points in the PtIDL1 peptide-treated plants. In contrast, the PtIDA peptide continued to enhance the percentage of leaf abscission up to 20% from 14 DAD to 18 DAD. The abscission enhancing effect of the PtIDA peptide decreased at 20 DAD, and by 22 DAD there was no difference in the percentage of leaves lost between the controls and the peptide-treated plants. Furthermore, plants treated with the PtIDA peptide lost their leaves significantly earlier than the control or PtIDL1-treated plants (Figure 1b). The results indicate that both the PtIDA and the PtIDL1 peptides can enhance dark-induced leaf abscission of *Populus*, but their effects are not the same, with the PtIDA peptide being more efficient than the PtIDL1 peptide. 

### 2.2. Exogenous Treatments of the *EgIDA* Peptide and Effects on Cell Separation in the Oil Palm Fruit AZ

Fruit abscission of oil palm occurs naturally with ripe fruit in the field and is induced by ethylene [19,20,21,22]. In a previous study, it was shown that at least two *IDL* transcripts (*EgIDA2* and *EgIDA5*) were expressed in the ripe fruit AZ, and this expression was enhanced by ethylene [17]. We developed an in vitro phenotype test to examine the abscission behaviour and to screen for individuals of oil palm that have delayed abscission [23,24]. We used this experimental system to examine the effect of the EgIDA5 peptide on oil palm ripe fruit abscission. Our hypothesis was that the synthetic EgIDA5 peptide could enhance the cell separation process of the ripe fruit AZ, which would provide further evidence that the IDA–HAE–HSL2 pathway functions during oil palm ripe fruit abscission. In previous experiments, it was found that ethylene induced fruit abscission, but the effect was more pronounced with ripe fruit [21]. In the current experiment, fruit from bunches at two different stages of development, approximately 120 and 150 days after pollination (DAP), were used to examine the effect of the synthetic EgIDA peptide on cell separation in the AZ. For these experiments, an abscission index (AI) was calculated based on four different phenotypic responses observed as described in the Materials and Methods. For the first biological repetition, AI variation depended significantly on the EgIDA peptide treatment (p = 0.000036), the age of the fruit (p = 0.00000), and the EgIDA × age of the fruit interaction (p = 0.000077). The EgIDA peptide had a positive effect on the AI at either concentration used. However, the effect of the peptide differed depending on the age of the fruit (Figure 2a). This indicated that the EgIDA peptide only had a significant effect on cell separation (more positive AI) of fruit at 150 DAP. These results were confirmed in a second biological repetition (Figure 2b). It was concluded that the EgIDA5 peptide enhanced abscission in ripe fruit, at a time when ethylene is being produced, but not in fruit at stages of development when ethylene production is low [25]. 

## 3. Discussion

The IDA–HAE–HSL2 organ abscission regulatory pathway began to be elucidated in transgenic antisense plants with a reduction of function of HAE, an LRR-RLK, in *Arabidopsis* [26]. Reduction of HAESE function resulted in a delay of floral organ abscission, with the severity of the phenotype correlated to the amount of HAE protein present. The results suggest that HAE may bind to an as yet unidentified ligand. The next piece of the puzzle came with the identification of ethylene-sensitive *ida* mutants that have a non-shedding floral organ phenotype [11]. The fact that *IDA* is a gene that encodes a small protein suggests it could be processed post-translationally to give rise to a peptide ligand recognized by a receptor, which together may act in signalling to control floral organ abscission in *Arabidopsis*. A 12 amino acid conserved motif named PIP in the C-terminal region of the IDA protein was thought to be important for ligand receptor function [11]. Further evidence came from a series of experiments that provided indirect evidence that HAE is the receptor for IDA [12]. That study demonstrated that the functional part of the IDA protein is found in the EPIP peptide, and that when flowers were exposed to 10 µM of the EPIP peptide they promoted up to 20% more floral organ abscission than untreated flowers. It was then shown that the dodecamer PIP peptide at nanomolar concentrations is able to activate HSL2 in transient tobacco cell assays [13]. Finally, studies revealed that the PIP peptide binds to a hormone-binding pocket in the receptor HAE, and that a central hydroxyproline residue in the PIP peptide anchors IDA to the receptor [27].

In the present study, we addressed whether the functional portion of IDA peptides, the PIP motif, could be bioactive to enhance the organ abscission processes of two widely divergent species, including leaf abscission of *Populus* and fruit abscission of oil palm. In both experimental systems used, the results indicate that the IDA peptides are bioactive and capable of enhancing both leaf and ripe fruit abscission in *Populus* and oil palm, respectively. Previous reports with citrus and litchi have provided evidence for the conservation of the IDA–HAE–HSL2 pathway in these species [15,16]. In citrus, overexpression of the citrus *IDA* gene *CitIDA3* in *Arabidopsis* results in early floral organ abscission and an increased AZ size, similar to the phenotype observed when the *Arabidopsis IDA* gene *AtIDA* is overexpressed in *Arabidopsis* [15,28]. In addition, overexpression of *CitIDA3* in the *ida-2* mutant rescues the abscission defect in these plants, providing further evidence of conservation of function [15]. With litchi, overexpression of the *IDL* gene *LcIDL1* in *Arabidopsis* also results in premature floral abscission [16]. As with citrus, the overexpression of the *IDL* gene *LcIDL1* in the *ida-2* mutant results in reversion to wild-type-like floral organ abscission. 

While both of these studies provided evidence of functional conservation using the *Arabidopsis* floral organ abscission system as a platform, neither of these studies addressed whether the IDA peptides have bioactivity to promote abscission, and whether the IDA–HAE–HSL2 pathway functions during organ abscission in either of these plant species. In addition, a recent study with *Lupinus luteus* L. revealed that exogenous treatment with EPIP peptides significantly increased the rate of flower abortion, which suggests that the IDA–HAE–HSL2 pathway functions in that system [29]. In the current study, we showed that the bioactivity of IDA peptides was sufficient to enhance both leaf and fruit abscission in *Populus* and oil palm, respectively. In addition, the experiments showed that a low concentration (0.1 µM) of peptide was sufficient for abscission-promoting bioactivity, similar to the effect of *Arabidopsis* IDA PIP observed in activation and binding studies [13]. The results suggest that the PIP peptide of IDA has the capacity to function as a peptide hormone in planta, and provides evidence that the IDA–HAE–HSL2 pathway functions in these two diverse organ abscission systems.

Several studies have revealed that exogenous ethylene treatments induce oil palm fruit abscission, but the enhanced effect is observed more clearly with ripe fruit [21,22,30]. In addition, at least two *EgIDA* transcripts (*EgIDA2* and *EgIDA5*) are expressed in the AZ during ripe fruit abscission of oil palm, and this expression is enhanced by ethylene [17]. In the current study, we found that the enhancing effect of the EgIDA peptide depended on the stage of development, with cell separation in the ripe fruit AZ being more sensitive to the treatments. In a recent literature review that addressed the position of the IDA–HAE–HSL2 pathway in the regulation of floral organ abscission in *Arabidopsis*, it was concluded that the pathway is downstream of ethylene and functions at the last stages of the abscission process [10]. Our data support this view given that the EgIDA peptide only enhances abscission of ripe fruit, at stages when ethylene is being produced in the fruit mesocarp [25].

Remarkably, despite the phylogenetic divergence of the two species and *Arabidopsis* in which the IDA–HAE–HSL2 pathway was discovered, and the difference in the organ abscission processes examined, including shade-induced leaf abscission of poplar, ripe fruit abscission of oil palm, and floral organ abscission of *Arabidopsis*, the PIP peptide of IDA enhanced cell separation in all systems. Our results suggest that these peptides, as well as the amino acids in the receptors that interact with the ligand peptide, according to the structure of IDA with HAE, are extremely well conserved.

## 4. Materials and Methods

### 4.1. Plant Material

Oil palm individuals were cultivated at the Centre de Recherches Agricoles – Plantes Pérennes (CRA-PP), Station de Pobè, Bénin and derived from a self-pollinated individual (DA115D) that originated from the Deli population. Eight to 12 fruit spikelets were removed from the upper middle section of each fruit bunch and used for the in vitro abscission test described below. The date of pollination was recorded in the field and used to define the developmental stages based on days after pollination (DAP) to select fruit material for the experiments.

For dark induction experiments, the hybrid aspen clone T89, *Populus tremula* L. X *P. tremuloides* Michx, was employed. Saplings were propagated in vitro before transfer to soil and grown in a glasshouse, as described in detail in [18].

### 4.2. In Vitro Phenotype Tests of Oil Palm Fruit Abscission

The abscission index (AI) was calculated from the absolute values obtained by an in vitro phenotype test previously described with modifications [24]. From the fruit spikelets collected as described above, approximately 24 similarly ripe fruit were selected and the base of each fruit containing the AZ was then sliced longitudinally to give at least two slices approximately 1–1.5 mm in length (Appendix A). This allowed each slice to contain both the large primary AZ and the adjacent AZs previously described [19,22]. A total of 45 fruit base slices from each bunch were collected and placed on Petri plates (15 slices per plate) with dampened Whatman filter paper supplemented with 2 mL of EgIDA5 synthetic peptide, which consists of the PIP motif (PIPPSGPSKRHN) synthesized and quantified as previously described [13,17], in solution at concentrations of 0, 0.1, and 1 µM. In addition, 30 µL of EgIDA5 peptide solution was deposited on the AZ of each slice (Appendix A). The fruit bases containing the AZ were then incubated at room temperature for 24 h. For the phenotype test, pressure was applied with forceps by twisting the slices to determine whether separation had occurred, and whether it occurred at both the primary and secondary AZs. For the phenotypes, four classes were defined and attributed to each fruit slice tested: A, no separation; B, partial separation in primary AZ only; C, extensive separation in primary AZ only; and D, complete separation in both primary and adjacent AZs (Appendix A). To calculate the AI, we used the following formula: (nA*−3 + nB*−1 + nC*1 + nD*3)/(nA + nB + nC + nD), where n = total number of observations attributed per class. This formula gave a range of AI values from −3 to 3 [24]. For the EgIDA5 peptide dose response experiments, fruit from bunches at around 120 and 150 days after pollination (DAP) were used. The effect of the EgIDA peptide and the age of fruit bunch were tested on the AI, using an analysis of variance (ANOVA) with two fixed classification criteria (IDA, age). In the event of a significant effect with more than two modes, which is the case with significant interactions, the Newman and Keul multiple comparison of means test was used [31,32].

### 4.3. Dark-Induced Abscission Experiments 

To induce abscission, blades of fully expanded leaves with a petiolar angle between 75° and 90° from 2 m tall trees were covered with aluminum foil. Either 30 µL of 0.1 µM of the synthetic peptide PtIDA or PtIDL1, which consists of the PIP motif (PtIDA, PIPPSGoSKRHN and PtIDL1, PVPPSAoSKRHN, where “o” stands for hydroxyproline; synthesized as previously described [13]), dissolved in water was applied to leaf axils. Leaf axils were treated at day 0 (beginning of dark induction) and day 7. For each treatment, four replicate trees with eight treated leaves were used. Abscission was scored on a daily basis by counting dropped leaves after gently shaking the trees.

## Figures and Tables

**Figure 1 plants-08-00143-f001:**
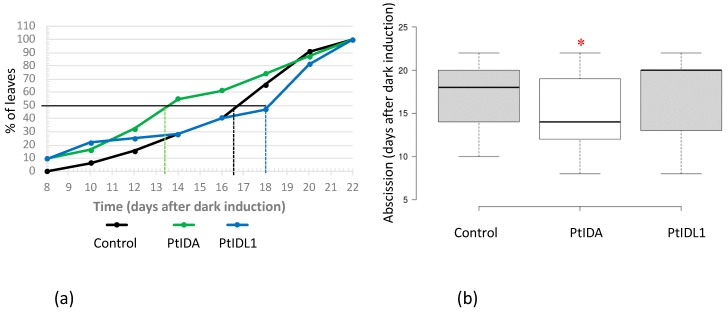
Poplar leaf abscission induced by dark treatment is enhanced in response to PtIDA or PtIDL1 peptide treatments of leaf axils. (**a**) Time course of treatment with 0.1 µM PtIDA, PtIDL1, or water (control) 7 days after dark induced abscission. The percentage of dark-induced leaves dropped. n = 31 leaves for PtIDA and n = 32 for control treatment (water) and PtIDL1; dotted lines represent timepoints when 50% of the leaves were dropped. (**b**) PtIDA-treated leaves separated significantly earlier (*, t-test; n = 31 or 32 leaves; p < 0.05) than mock-treated leaves. Boxplot: central lines represent the median, limits of boxes are the 25th and 75th percentiles, and whiskers are the 1.5× interquartiles.

**Figure 2 plants-08-00143-f002:**
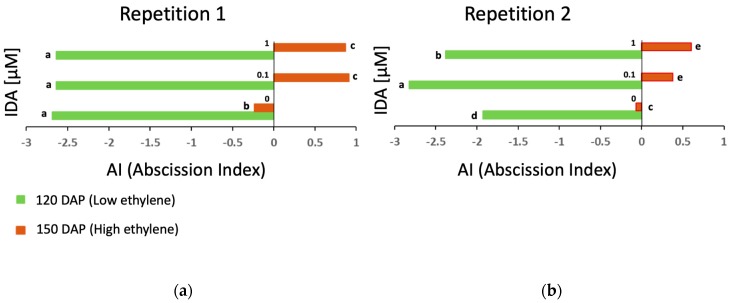
Cell separation in the oil palm fruit abscission zone (AZ) is enhanced significantly by treatments with the EgIDA peptide at concentrations of 0.1 or 1 µM, and the effect is only observed with ripe fruit. (**a**) There is a significant (p = 0.000036) effect of the EgIDA peptide on the abscission index (AI), and the effect is age-dependent (p = 0.00000). (**b**) A second repetition confirmed the significant (EgIDA × age of the fruit interaction, p = 0.000824) effect of the EgIDA peptide to enhance separation in AZs of ripe fruit (150 days after pollination (DAP)) when ethylene is produced, and not at an earlier stage of development (120 DAP) when ethylene production is low. Different lower-case letters represent statistically significant differences. Negative AI means less abscission, positive AI more abscission. The AI was calculated as described previously [24].

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
