# Peer review of "The PIP Peptide of INFLORESCENCE DEFICIENT IN ABSCISSION Enhances Populus Leaf and Elaeis guineensis Fruit Abscission"

_plants, 2019, doi:10.3390/plants8060143_

Round 1
Reviewer 1 Report
The authors have addressed the concerns from previous review process, and I believe it is now ready for publication.
Author Response
Thank you for your positive remarks and review.
Reviewer 2 Report
The authors responded to my previous comments. The manuscript is well written and the work well described. I don't have any further remarks.
Author Response

(The authors gave the same response as above.)

Reviewer 3 Report
This manuscript describes two main experiments where the PIP from IDA was applied to two different species and a change in abscission characteristics were noted. I have a few concerns on the manuscript as indicated below.
In hybrid aspen, the authors used a dark-induced abscission system to test IDA effects. It would have been useful to determine the same in non-induced conditions as well to see if the peptide is able to elicit abscission responses in mature AZs.
The authors indicate that previous studies reported induction of IDA transcripts during dark-induced abscission. It would be useful to compare the timing of that transcript induction with the acceleration of abscission responses noted here. Further, how does this relate to the induction timing of other transcripts such as HAE?
The authors discuss an abscission index for their study on oil palm. It is much later in the materials that the description of this index is provided. This was difficult to follow. A brief description of the index would be useful when the results are being presented.
It was very unclear to me about how the PIP was made and quantified. This is not explained in the materials and methods and this information needs to be provided.
Author Response
Response to Reviewer 3 comments
· “In hybrid aspen, the authors used a dark-induced abscission system to test IDA effects. It would have been useful to determine the same in non-induced conditions as well to see if the peptide is able to elicit abscission responses in mature AZs. The authors indicate that previous studies reported induction of IDA transcripts during dark-induced abscission. It would be useful to compare the timing of that transcript induction with the acceleration of abscission responses noted here. Further, how does this relate to the induction timing of other transcripts such as HAE?”
· We agree with the reviewer that these would have been a very good experiments to perform, however, to perform a new set of experiments now would be very difficult under the circumstances, mainly because of a professional career change (i.e. new job) for the primary investigator that performed the original experiments with Poplar. I hope that the Reviewer will understand this situation and that the lack of results from the proposed extra experiment does not take away from the solid data obtained from the experiments presented in the manuscript.
· “The authors discuss an abscission index for their study on oil palm. It is much later in the materials that the description of this index is provided. This was difficult to follow. A brief description of the index would be useful when the results are being presented.”
· Thank you for this improvement. A brief description was added to the results as the data are presented.
· “It was very unclear to me about how the PIP was made and quantified. This is not explained in the materials and methods and this information needs to be provided.”
· Thank you again. The clarification of the synthesis and quantification of these synthetic peptides was indicated in the Materials and Methods section by referring to the methods used in a previous study, and a clarification was also added to the Results section to indicate that these were synthetic peptides used for the study.
This manuscript is a resubmission of an earlier submission. The following is a list of the peer review reports and author responses from that submission.
Round 1
Reviewer 1 Report
The manuscript "The IDA peptide signaling pathway functions during Populus leaf and Elaeis guineensis fruit abscission" is interesting and shows the peptide treated phenotypes.
Few thoughts for the authors to consider.
Only the phenotypes themselves are not sufficient to draw the conclusion of the entire signing pathway, more supporting evidence is required.
It may be a good idea to have some significant tests on the data analysis.
Can the authors justify why choosing leaf abscission in one species and fruit abscission in another? How about both processes, are they performing similarly?
Reviewer 2 Report
The manuscript of Tranbarger et al, entitled "The IDA peptide signalling pathway functions during Populus leaf and Elaeis guineensis fruit abscission", presents evidence of an active IDA-HAE-HSL2 signalling pathway in poplar and oil palm by measuring the effect of IDA peptide application on leaf and fruit abscission. While previous studies were based on testing the activity of the genes coding for the IDL, HAE, HSL2 from other species by expressing them in Arabidopsis, this is the first studies to test for the activity of the peptide for promoting abscission in leaves of poplar and fruit of oil palm.
The experiments are well designed and results are solid. I would have only minor comments.
Concerning Fig1, it appears that there might be a formatting problem with the line marking the 50% leaf abscission. It is now dropped to 42%, with a shift on the corresponding time intersection for each treatment. Also, some explanation of the graph b would be useful: do you show mean or median value, to what corresponds to the values of the boxes etc. Please mark the significant difference on the graph.
Was any statistics applied to the results in Figure 2?
One other minor English error in line 77: "similar genes that encode for IDL...", please use "code for" or "encode IDL"
Reviewer 3 Report
This manuscript is the first to report a direct promoting effect of the signaling peptide IDA on abscission in other species than Arabidopsis. Orthologues of IDA and its receptors have been identified previously in allorders of flowering plants, and shown to be expressed in the fruit and leaf abscission zones of, respectively, so different species as oil palm and poplar. This is an important finding with regards to the question of how similar the molecular mechanisms governing abscission are in different plants and in different organs. In connection to the Special Issue on cell separation, this is highly relevant, especially since so divergent Plant and organs are used.
Minor comments:
Lines 123: reference to Supplementary figure here?
Line 64-66:The authors should check their references;, reference should go to Stenvik et al 2008 (which is referred to later as ref.27), rather than Stenvik et al 2006. Furthermore reference to Butenko et al, Plant Cell 2014, would be appropriate.
This paper demonstrates binding and activation of HSL2 by IDA, and shows that the most efficient peptide is the 12 amino acid long PIP peptide with hydroxylation of a central proline residue.
Lines 165-170: This is a bit confusing, here again reference to the activity of the PIP peptide published in Butenko et al, Plant Cell 2014, would be appropriate. The solved structure of the IDA peptide bound to the HAE receptor could also be referred to (Santiago J, Brandt B, Wildhagen M, Hohmann U, Hothorn LA, Butenko MA, Hothorn M.Elife. 2016 Apr 8;5. pii: e15075.
Line 159: it should be ethylene insensitive, not ethylene sensitive
Line 191: the effect is close to the effect of IDA PIP in activation and binding studies shown in Butenko et al 2014
.
The poplar figure should be checked